# Advancing data to care strategies for persons with HIV using an innovative reconciliation process

**Merceditas Villanueva** [1]*, **Janet Miceli**[1¤a], **Suzanne Speers**[2], **Lisa Nichols**[1], **Constance Carroll**[1¤b], **Heidi Jenkins**[2], **Frederick Altice**[1]

**1** Department of Internal Medicine, Section of Infectious Disease, HIV/AIDS Program, Yale University School of Medicine, New Haven, CT, United States of America, **2** CT Department of Public Health, Hartford, CT, United States of America

¤a Current address: Yale Program on Aging, New Haven, CT, United States of America
¤b Current address: Yale Center for Clinical Investigation, New Haven, CT, United States of America
* merceditas.villanueva@yale.edu

## Abstract

### Background

UN AIDS has set ambitious 95-95-95 HIV care continuum targets for global HIV elimination by 2030. The U.S. HIV Care Continuum in 2018 showed that 65% of persons with HIV (PWH) are virally suppressed and 58% retained in care. Incomplete care-engagement not only affects individual health but drives ongoing HIV transmission. Data to Care (D2C) is a strategy using public health surveillance data to identify and re-engage out-of-care (OOC) PWH. Optimization of this strategy is needed.

### Setting

Statewide partnership with Connecticut Department of Public Health (CT DPH), 23 HIV clinics and Yale University School of Medicine (YSM). Our site was one of 3 participants in the CDC-sponsored RCT evaluating the efficacy of DPH-employed Disease Intervention Specialists (DIS) for re-engagement in care.

### Methods

From 11/2016-7/2018, a data reconciliation process using public health surveillance and clinic visit data was used to identify patients eligible for randomization (defined as in-Care for 12 months and OOC for subsequent 6-months) to receive DIS intervention. Clinic staff further reviewed this list and designated those who would not be randomized based on established criteria.

### Results

2958 patients were eligible for randomization; 655 (22.1%) were randomized. Reasons for non-randomizing included: well patient [499 (16.9%)]; recent visit [946 (32.0%)]; upcoming visit [398 (13.5%)]. Compared to non-randomized patients, those who were randomized

**Funding:** This work was supported by the Centers for Disease Control and Prevention research grant FOA PS14-001. The primary recipient of this award was HJ; sub recipients were MV and FA. The sponsor was involved in the study design as part of the overall grant RCT. Yale University was involved in the data collection, analysis and manuscript preparation.

**Competing interests:** The authors have declared that no competing interests exist.

were likely to be younger (mean age 46.1 vs. 51.6, p < .001), Black (40% vs 35%)/Hispanic (37% vs 32.8%) [(p < .001)], have CD4<200 cells/ul (15.9% vs 8.5%, p < .001) and viral load >20 copies/ml (43.8% vs. 24.1%, 0<0.001). Extrapolating these estimates to a statewide HIV care continuum suggests that only 8.3% of prevalent PWH are truly OOC.

## Conclusions

A D2C process that integrated DPH surveillance and clinic data successfully refined the selection of newly OOC PWH eligible for DIS intervention. This approach more accurately reflects real world care engagement and can help prioritize DPH resources.

## Introduction

Antiretroviral therapy (ART) can durably suppress HIV leading to individual and public health benefits. The Joint United Nations Programme on HIV/AIDS (UNAIDS) has set 95-95-95 goals (95% of persons with HIV (PWH) diagnosed, 95% of those diagnosed on ART, and 95% of those on ART with viral suppression) for 2030 [1]. These targets have been adopted by many countries to assess progress towards public health goals and to guide resource allocation. These targets are based on the HIV Care Continuum model to depict population level success according to stages of receipt of care [2]. In 2018, the United States Centers for Disease Control (CDC) estimated that 58% of persons with HIV (PWH) are retained in care and 65% have achieved viral suppression (<200 copies/ml on the most recent viral load test) [3]. PWH not retained in medical care and/or not virally suppressed are estimated to account for >60% of HIV transmissions in the United States [4, 5]. Existing strategies to promote retention in care and viral suppression [6, 7] are suboptimal.

Public health departments have the potential to play an important role in achieving these targets. In the U.S., interventions that incorporate health department (HD) and clinic partnerships define "Data to Care (D2C)," [6–8] a public health strategy that uses HIV surveillance and other data sources to identify and link newly diagnosed or to re-engage out-of-care (OOC) PWH [9]. The CDC's HIV/AIDS Prevention Strategic Plan advocates expanding D2C programs, targeting PWH who have fallen out of or never entered care in order to improve HIV viral suppression [10]. While this approach has garnered support in the U.S., the logistics of implementation are not well established and varying algorithmic approaches are challenging to adopt. This stems partly from evolving definitions of being in care; as ART has improved, recommendations for frequency of visits and HIV laboratory monitoring have changed. The current U.S. Department of Health and Human Services (DHHS) guidelines recommend that after 2 years on ART with consistently suppressed viral load (VL), persons with CD4 count 300–500 cells/mm3 should have CD4 count monitoring every 12 months and those with CD4 count >500 cells/mm3 can have optional CD4 count monitoring [11]. For persons with stable immunologic status and VL <200 copies/ml for 2 years, VL monitoring can be extended to every 6 months. Thus, definitions of being engaged in care based on VL monitoring can be met at minimum with VL testing every 6 months.

Furthermore, determining who is out-of-care and would benefit from a DPH intervention requires refining the operational definition. Historically, retention in care, commonly defined as 2 visits to an HIV prescribing provider at least 90 days apart over 12 months, is determined retrospectively. D2C is a prospective strategy that requires implementation of practical re-engagement interventions after determining who is OOC. Healthcare provider-initiated

interventions (e.g. text messaging appointment reminders, enhanced case management, counseling/behavioral modification) have had modest effectiveness [12, 13]. HD-initiated re-engagement projects have included technology-focused approaches [14–16] and use of HD staff to review surveillance records from selected healthcare facilities to guide contact tracing of OOC patients [17, 18]. Alternatively, clinic-initiated, surveillance-informed approaches where OOC lists were generated by clinics and modified by HD surveillance staff have been used to guide clinic-re-engagement interventions [19–21].

Project CoRECT (Cooperative Re-Engagement Clinical Trial) was a CDC-funded randomized controlled trial (RCT), involving the identification of OOC PWH using HD surveillance and clinic-generated data, with randomization of eligible PWH to a Disease Intervention Specialist (DIS) vs. clinic standard of care (SOC). Primary outcomes were re-engagement at 90 days, retention in care at 12 months and viral suppression at 12 and 18 months. Connecticut (CT) was one of 3 sites involved (other sites were Philadelphia and Massachusetts). We describe and validate our local D2C algorithm which involved a bi-directional data sharing process between the CT Department of Public Health (DPH) and 23 participating HIV clinics that illustrates how refinement of the D2C process using specific data reconciliation strategies can more accurately inform real-world approaches to re-engagement in care.

## Methods

### Study background

CT is a small state with a 2018 HIV prevalence of 295.7/100,000. The CT DPH partnered with Yale University School of Medicine to conduct the study. Overall, 23 clinics ("CoRECT clinics") which are estimated to account for >95% of PWH in CT participated. Communication between the CT DPH (HIV Epidemiologist) and CoRECT clinics was facilitated by Yale University study staff. Study recruitment occurred from 11/2016-7/2018.

### Protocol for defining out of care and randomization

Eligible patients (defined by CDC) included those who received HIV care for 12 months and then disengaged as defined by no clinic visit with an HIV prescriber for 6 months and/or no CD4/Viral Load (VL). Study enrollment design is illustrated in S1 Fig.

### Statistical methods

Data analysis was performed using SAS software, version 9.4 of the SAS System for Windows. Categorical variables were described using frequencies and percentages; continuous variables were characterized as means with standard deviations. Pearson's chi-squared test was utilized to analyze frequency distributions; the Student's T test was used for analyzing continuous variables.

### Ethics statement

The Yale University Institutional Review Board determined that this study was not considered to be Human Subjects Researcfh and did not require IRB review (IRB/HSC# 1510016672)

## Results

### Patient enrollment

Overall, 2958 patients were "potentially OOC" and eligible for case conference (Fig 1). Of these, 763 (25.8%) were in Box B; 1,342(45.4%) in Box C; 853 (28.8%) in Box D. There were 655 randomizable patients: 333 (50.8%) to DIS and 322 (49.2%) to SOC.

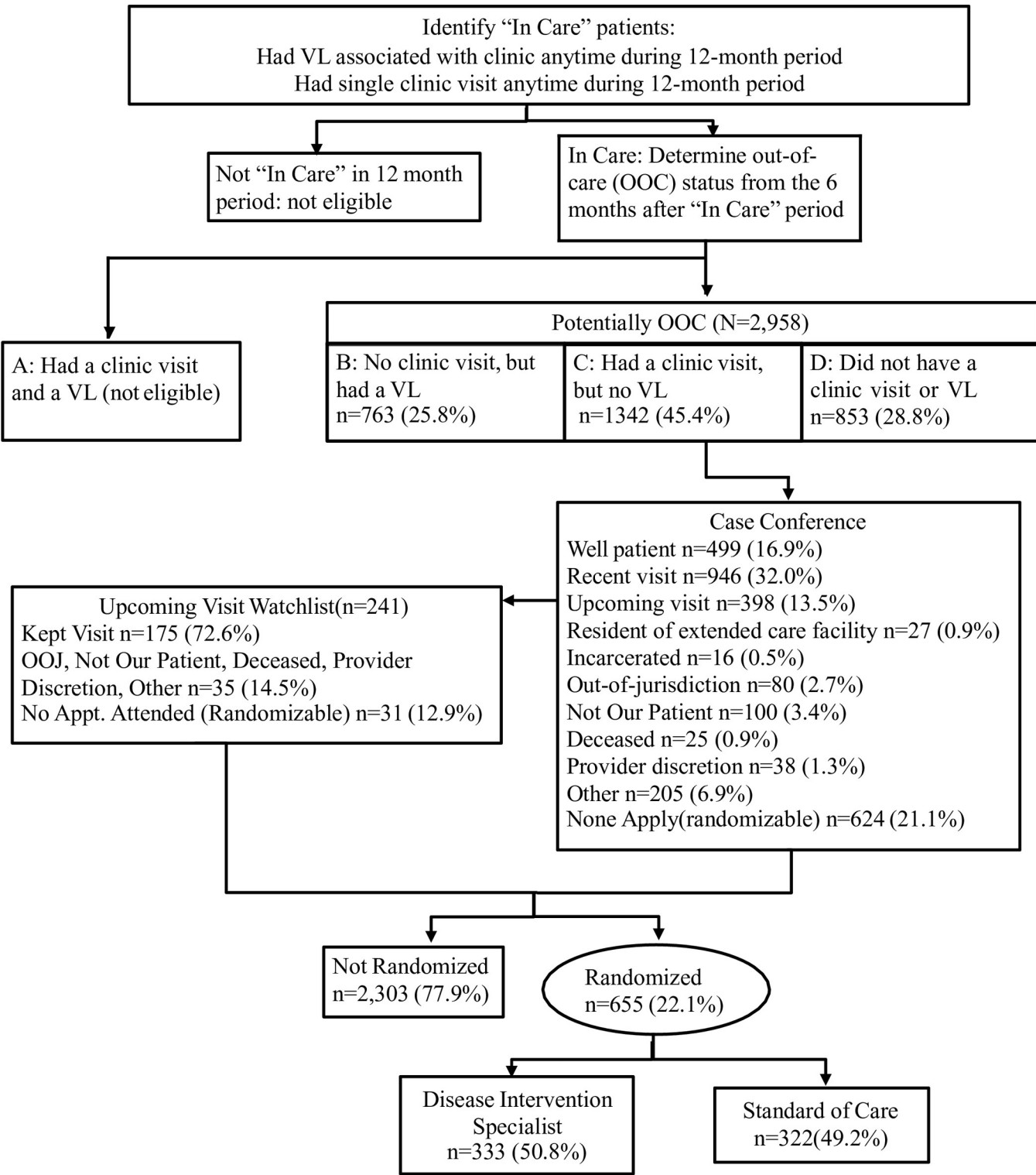

**Fig 1. Flowchart for enrollment of patients who disengage from medical care.** This figure shows the algorithm used by CoRECT clinics and the DPH to identify PWH who were in care, out of care and those eligible for randomization to the DIS intervention. There were 2,958 patients who were potentially out of care, divided up into Box B, C, D based on clinic and VL data during the 6 month OOC period. Case conferencing outcomes delineating patient care status including those placed into an upcoming visit watchlist are shown. Ultimately, 655 patients were randomized to DIS vs SOC.

Among potentially OOC patients undergoing case conference, the major dispositions included: 499 (16.9%) well patients, 946 (32.0%) recent visit, 398 (13.5%) upcoming visit. From this initial review, 624 (21.1%) were eligible for randomization. Of 241 patients from the Upcoming Visit watchlist, 31 (12.9%) did not keep their appointment and were eligible for randomization.

## Comparison of randomized vs non-randomized patients

Table 1 compares demographics of randomized vs non-randomized patients. Overall, patients who were potentially OOC had a mean age of 50.3 years; 64.7% were male; 33.7% were Hispanic, 36.2% were Black; 27.6% White. Among randomized patients (N = 655), mean age was 46.1 years; 62.4% were male; 37% were Hispanic, 40.3% were Black, 20.8% white. Randomized patients were younger (randomized vs. non-randomized under 30, 13.9% vs 6.4%; age 30–39, 18% vs. 11.7%) (p<0.001), mean age 46.1 vs 51.6, p<0.001); had higher proportions of persons of color (Black (40.3% vs 35%) or Hispanic (37% vs 32.8%), p<0.001). Among those randomized, the distribution was: Box B 124 (18.9%); Box C 228 (34.8%); Box D 303 (46.3%); this distribution was clinically significantly different compared to non-randomized (p < .001).

**Table 1. Demographics of randomized vs non-randomized out of care PWH.**

| Characteristics | Total N = 2,958 | Not Randomized n = 2,303 | Randomized n = 655 | P Value |
|---|---|---|---|---|
| Age mean (median), y | 50.3 (52.2) | 51.6 (53.1) | 46.1 (47.3) | < .001 |
| Age group, y, No. (%) | | | | < .001 |
| Under 30 | 239 (8.1) | 148 (6.4) | 91 (13.9) | |
| 30–39 | 388 (13.1) | 270 (11.7) | 118 (18.0) | |
| 40–49 | 618 (20.9) | 443 (19.2) | 175 (26.7) | |
| 50–59 | 1,090 (36.9) | 908 (39.4) | 182 (27.8) | |
| Over 60 | 623 (21.1) | 534 (23.2) | 89 (13.6) | |
| Sex at Birth, No. (%) | | | | 0.18 |
| Male | 1,913 (64.7) | 1,504 (65.3) | 409 (62.4) | |
| Female | 1,045 (35.3) | 799 (34.7) | 246 (37.6) | |
| Race, No. (%) | | | | < .001 |
| Hispanic | 997 (33.7) | 755 (32.8) | 242 (37.0) | |
| Black, Not Hispanic | 1,070 (36.2) | 806 (35.0) | 264 (40.3) | |
| White, Not Hispanic | 815 (27.6) | 679 (29.5) | 136 (20.8) | |
| Other | 76 (2.6) | 63 (2.7) | 13 (2.0) | |
| Exposure Category, No. (%) | | | | 0.07 |
| MSM | 871 (29.5) | 677 (29.4) | 194 (29.6) | |
| IDU | 837 (28.3) | 659 (28.6) | 178 (27.2) | |
| Heterosexual Only | 904 (30.6) | 714 (31.0) | 190 (29.0) | |
| MSM & IDU | 78 (2.6) | 55 (2.4) | 23 (3.5) | |
| Other | 62 (2.1) | 40 (1.7) | 22 (3.4) | |
| None Identified or Reported | 206 (7.0) | 158 (6.9) | 48 (7.3) | |
| OOC Classification, No. (%) | | | | <0.001 |
| Box B(No Clinic Visit, VL) | 763(25.8) | 639(27.8) | 124(18.9) | |
| Box C(Clinic Visit, No VL) | 1,342(45.4) | 1,114(48.4) | 228(34.8) | |
| D(No Clinic Visit & No VL) | 853(28.8) | 550(23.9) | 303(46.3) | |

Because of rounding, percentages may not total 100. MSM, men who have sex with men; IDU, injection drug user; OOC (out of care); VL, HIV viral load.

### HIV clinical status of potentially OOC PWH

The last available in-care CD4 count and VL between randomized and non-randomized patients are shown in Table 2. Overall, 1578 (59.5%) had CD4>500 cells/ul and 2,108 (71.5%) has VL < 20 copies/ml). A comparison of randomized vs non-randomized showed CD4 <200 cells/ul [95 (15.9%) vs 175 (8.5%), p < .001] and VL >20 copies/ml [43.8% vs. 24.1%, p < .001]. There were no significant differences in CD4 count distribution by Box classification (Table 3). Box D patients had the highest proportion of detectable VL (Box D 269 (31.7%) vs Box B 224 (29.4%) vs Box C 347 (25.9%), p = 0.01).

### Effect on the HIV care continuum

We extrapolated our results to revise statewide estimates of retention in care as represented in the CT HIV Continuum of Care. Given our recruitment period, we applied revised estimates to the 2017 CT HIV Continuum of Care [28] which estimated that of 10,617 prevalent HIV cases, 6,616 (62.3%) were retained in care (2 VLs at least 3 months apart) leaving 4,001 (37.7%) that would be designated as OOC (see Fig 2). If we apply our above estimate (Fig 1) that 77.9% of potentially OOC are not randomizable (i.e. truly in care as they are well patients, had recent visit, etc), then an additional 3,117 cases (77.9% x 4001) would be added to those retained (N = 6,616), leading to a total of 9,733 (92%) estimated to be truly in care. Conversely, this leads to a revised cascade of care gap estimate of 884 (8.3%) as truly OOC (not retained).

The estimated additional re-classified in-care cases (accounting for an added 29.3% of total prevalent cases) can be atrributed to specific Box classifications based on calculated contributions of each Box to the non-randomized group as in Table 1 (specifically Box B 27.8%; Box C 48.4%; Box D 23.9%). The additional in-care cases (N = 3117) would be distributed as follows with respective contributions to prevalent cases (N = 10,617): Box B 866 (8.1%), Box C 1508 (14.2%), Box D 745 (7.0%).

## Discussion

Data to care (D2C) has been defined as a public health strategy that uses HIV surveillance to identify OOC PWH and thus optimize the HIV Care Continuum [9]. Although the CDC has emphasized the importance of D2C strategies in its 2017–2020 Strategic Plan [29] the approaches are highly variable. Project CoRECT is the first RCT to evaluate a combined Health Department and clinic-based D2C model aimed at improving re-engagement in care for PWH

**Table 2. Last in care CD4 and HIV viral load by randomization status.**

| In Care Lab Results | Total | Not Randomized | Randomized | P Value |
|---|---|---|---|---|
| CD4[a] group, n (%) | N = 2,652 | n = 2,056 | n = 596 | < .001 |
| < 200 | 270 (10.2) | 175 (8.5) | 95 (15.9) | |
| 200–299 | 211 (8.0) | 151 (7.3) | 60 (10.0) | |
| 300–499 | 593 (22.4) | 447 (21.7) | 146 (24.5) | |
| ≥ 500 | 1,578 (59.5) | 1,283 (62.4) | 295 (49.5) | |
| Viral Load[b] group, n (%) | N = 2948 | n = 2,295 | n = 653 | < .001 |
| Undetectable (≤ 20) | 2,108 (71.5) | 1,741 (75.9) | 367 (56.2) | |
| Detectable (> 20) | 840 (28.5) | 554 (24.1) | 286 (43.8) | |

[a] cells/µl.

[b] copies/ml.

Because of rounding, percentages may not total 100.

**Table 3. Last in care CD4 and HIV viral load by box classification.**

| In Care Lab Results | Total | Box B | Box C | Box D | P Value |
|---|---|---|---|---|---|
| CD4[a] group, n (%) | N = 2,652 | n = 664 | n = 1,214 | n = 774 | 0.13 |
| < 200 | 270 (10.2) | 81 (12.2) | 103 (8.5) | 86 (11.1) | |
| 200–299 | 211 (8.0) | 51 (7.7) | 98 (8.1) | 62 (8.1) | |
| 300–499 | 593 (22.4) | 142 (21.4) | 266 (21.9) | 185 (23.9) | |
| ≥ 500 | 1,578 (59.5) | 390 (58.7) | 747 (61.5) | 441 (57.0) | |
| Viral Load[b] group, n (%) | N = 2,948 | n = 762 | n = 1338 | n = 848 | 0.01 |
| Undetectable (≤ 20) | 2,108 (71.5) | 538 (70.6) | 991 (74.1) | 579 (68.3) | |
| Detectable (> 20) | 840 (28.5) | 224 (29.4) | 347 (25.9) | 269 (31.7) | |

[a] cells/μl.

[b] copies/ml

Because of rounding, percentages may not total 100.

Box B, No clinic visit;Box C, no viral load; Box D, no clinic visit or viral load during the out of care window.

who have fallen out-of-care. In this multi-site trial, each participating site (Philadelphia,MA and CT) used the same OOC definition which effectively targeted a "newly out of care" population (no visit or HIV lab for 6 months). Each of the sites varied in their implementation. In

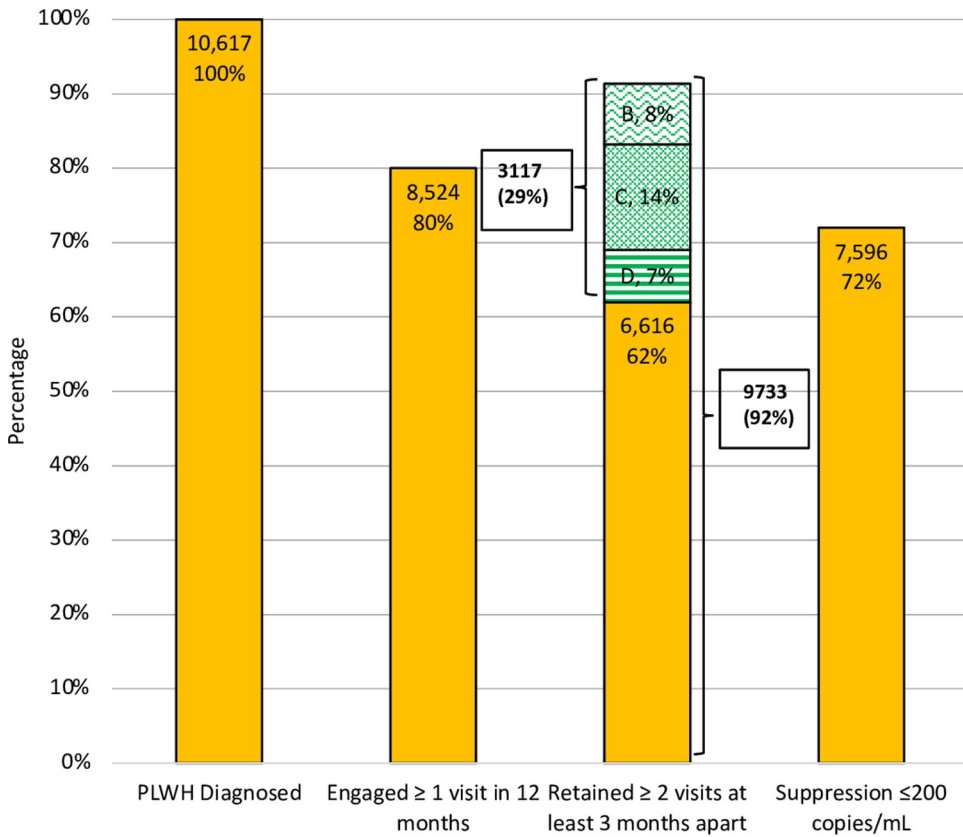

**Fig 2. Revised HIV continuum of care CT 2017.** The CT DPH publishes a yearly HIV continuum of care: (https://portal.ct.gov/DPH/AIDS—Chronic-Diseases/Surveillance/Connecticut-HIV-Statistics). In 2017, there were 10,617 diagnosed cases of which 6,616 (62%) were estimated to be retained in care. Applying revised estimates based on our findings that 77.9% of potentially OOC are not randomizable (i.e. considered in care), then an additional 3,117 cases would be added to those retained (N = 6616), leading to a total of 9,733 (91.7%) estimated to be in care and 884 (8.3%) truly OOC. The percentages of prevalent cases attributed to Box B, C, D are shown.

CT, we implemented a bidirectional data-exchange process based on simultaneously generated surveillance and clinic data with refinement of care status based on clinic-level assessment. This approach enabled more nuanced definition of a group of PWH deemed to be truly OOC and eligible to be randomized for enhanced case-finding by a Disease Intervention Specialist (DIS). We found that compared to non-randomizable PWH, this group was younger and included persons of color consistent with PWH historically more difficult to retain in care [30]. Those randomized also had a higher proportion with CD4 count <200 cells/μl and detectable HIV VL (>20 copies/ml), suggesting that our data reconciliation process correctly identified those with a demographic and virologic profile associated with being OOC.

Previously described D2C approaches have relied on Health Department (HD)- initiated identification of PWH presumed to be OOC using HIV Surveillance Registries to identify PWH presumed OOC followed by assignment to Field Services case workers for re-engagement [17, 18]. Subsequent studies have highlighted that surveillance data are not sufficient to fully capture those who are truly OOC and may result in inefficient case-finding efforts. In one study, OOC patients assigned to an Expanded Partner Services (ExPS) advocate found that only 23.7% of located cases were truly OOC, after re-classification of persons as deceased, moved out of jurisdiction or current to care [31]. Another study used extensive data searches (death registries, AIDS Drug Assistance Program, EMRs) as well as contacting of providers or patients and found that 28% of cases were truly OOC [32]. Our study corroborates these findings, highlighting that surveillance systems are often out-of-date and overestimate OOC numbers.

In an alternative approach, clinic-initiated efforts with HD input illustrate the same challenges with ascertainment of real-time care status of PWH. An example is a North Carolina program where clinics made referrals to State-employed Bridge Counselors (SBCs) with access to HIV surveillance data for those who had not kept medical appointments in 6–9 months and could not be contacted [20, 33]. In another study, a clinic-initiated list to define at-risk patients within Ryan White-funded clinics was "cross-checked" with HD surveillance and found that 36% were in care elsewhere, 8% returned to care on their own, and 29% were unable to be located [21]. Another study surveyed a subset of patients who were 210 days late for an HIV appointment and found that clinic and surveillance-based definitions of OOC were both inaccurate [34]. Bove *et al* used a similar approach and found that 79% of presumed OOC had moved or were out of jurisdiction [19]. Thus, clinic-initiated, surveillance-informed models may capture only persons for whom clinics are concerned and may miss a group of OOC persons that surveillance-generated approaches might capture [35].

More recent approaches such as the CDC-funded Partnerships for Care (P4C) demonstration projects aimed to create guidelines for data sharing by HDs and community health centers. In the Maryland Partnerships for Care (P4C) project [36], the MDH (Maryland Department of Health) matched eHARS to the EMRs of 4 participating health centers; patients without CD4 or VL data within 13 months or not virologically suppressed were categorized as not-in-care. MDH conducted case reconciliation conferences and found that only 7.5% of patients were actually not in-care. In the Massachusetts P4C project, the HD and 6 participating community health centers generated lists of patients not-in-care based on lack of CD4/VL and missed clinic visits over a 6-month time period. They also used a case reconciliation conference call to refine care status and to solicit provider concerns regarding re-engagement and found that only 5.9% of patients required public health field services intervention [37]. Our case reconciliation process had similar features but used a specific time frame emphasizing recently OOC based on a previous 12 month in-care period and was expanded to other HIV clinics beyond community health centers. Our application of these findings to a revised HIV

Continuum of care highlights that current published cascades overestimate the numbers of PWH who are OOC and have implications for HD and clinic resource allocation.

The D2C process is labor-intensive for both HDs and clinic staff with the latter group often not funded or trained to generate and reconcile lists of potentially OOC patients. One novel aspect of our study categorized patients based on absence of visits and/or VL which could enable further risk stratification. We found that the majority of potentially OOC fell into Box C (clinic visit, no VL in 6 months) possibly due to DHHS recommendations that stable patients do not need to get VLs drawn as frequently. Such patients may be considered "Current to Care," attending clinic but not getting frequent lab draws; this group tends to be clinically well [38]. In contrast, we found that patients in Box D (no VL or no clinic visit in 6 months) made up a large proportion of the randomized group and were more likely to have CD4 <200 cells/μl and VL >20 copies/mL. We believe that this may be useful for identifying a high-risk OOC group in need of more intensive re-engagement services, especially if automated systems from EMRs can generate lists of patients without clinic visits. We recognize that the additional step of case-conferencing between HDs and clinics to more accurately define OOC is time-consuming; however, if such efforts are concentrated on patients who meet the Box D classification, case-conferencing can be targeted to the sites with greatest number of patients who meet the criteria. Given limited resources in HDs, this could result in streamlining field worker (DIS) case-finding efforts [39].

The resources required for this D2C intervention are considerable with implementation costs in our 3 CoRECT sites varying from $14,145 to $26,058/month and labor hours from HD and clinic staff ranging from 224–650 hours/month [40]. It is possible that these hours may decrease as D2C processes become streamlined and routinized. Future work may define whether or not a 6-month out of care period is optimal. While most patients appeared clinically well in the group defined as potentially OOC (e.g. 71.5% had VL <20 copies/ml as their last in-care VL in our study), among the randomized group, 43.8% had VL >20 copies/ml, at their last in-care lab suggesting the potential benefit of early re-engagement.

Potential limitations to this approach include heterogeneity of clinic personnel's knowledge about individual patient clinical status. The relative lack of integrated data sources led to reliance on the ability of clinic staff with heterogeneous backgrounds to ascertain patient status without an independent verification.

## Conclusions

In conclusion, we used a bidirectional data exchange to reconcile HD surveillance and clinic data to refine the data-to-care process so that identification of PWH who have recently fallen out-of- care with linkage to a HD field worker becomes a manageable process. This evolution of the D2C process engages both HD and clinic staff and builds a collaborative environment that adds to the re-engagement toolbox of HIV clinics.

## Supporting information

**S1 Fig. Study enrollment design. STEP 1:** Data Exchange and Reconciliation (I→IV). *In Care and Out of Care Definitions*: "In care" patients were required to meet both of the following criteria: at least one clinic visit and an HIV VL within a designated 12-month period (I→II). "Out-of-care" patients had either no clinic visit or no VL during the subsequent six months (II→III). *Data Exchange Process*: Clinics and DPH transferred client-level data via a secure file transfer mechanism meeting the federal government's security compliance requirements. Each clinic was responsible for sending data files to the DPH. The designated out-of-care period would end one month prior to when the request for data was sent, to accommodate a

'lag period' (III→IV) due to potential delays in HIV lab reporting. Clinics submitted two lists: (1) patients who attended a clinic visit within the 12-month date range (2) patients with no clinic visit in the subsequent 6-months. Clinic data sources included CAREWare (an electronic health system developed by the Health Resources and Service Administration (HRSA) for Ryan White Grant recipients) and Electronic Medical Record (EMR). The DPH matched data from the clinics with HIV VL data from eHARS (enhanced HIV/AIDS Reporting System), the HIV Surveillance system. Patients were further classified based on clinic visit attendance and VL data (Fig 1). Patients with a clinic visit and VL in the 6-month timeframe were excluded (Box A); the remainder were designated as "Potentially OOC" and subdivided as: Box B (no clinic visit, had VL); Box C (had clinic visit, no VL); Box D (no clinic visit, no VL). **STEP 2**: Case Conference (IV→V). The matched list of potentially OOC was sent back to the clinics to designate a disposition based on review of EMRs and discussion with other clinic staff. Dispositions included the following in stepwise order at time of case conference: (1) Well Patient (2 consecutive undetectable VL of ≤20 copies/ml at least 6 months apart and no evidence of detectable VL during the "in care" or "out of care" period (2) Recent Clinic Visit (within the lag period) (3) Upcoming Visit (scheduled within 3 months of lag period) (4) Resident of Extended Care Facility (5) Incarcerated (6) Moved Out of State (7) Not Our Patient (transferred to another HIV clinic) (8) Deceased (9) Provider Discretion (provider concerns for privacy or other health issues) (10) Other. Patients who did not meet any of these criteria were randomizable. In addition, patients who were initially designated as having an Upcoming Visit were put on a Watchlist (see Fig 1) and were subsequently reviewed to confirm if the clinic visit was kept; those who did not attend their visit were randomized. The case conference process involved communication between the clinics and the DPH staff, usually by telephone and on an *ad hoc* basis as needed by the clinics. Yale facilitated the process by working with the DPH epidemiologist to pre-identify non-randomizable patients (e.g. well patients (based on (VL <20 copies/ml), deceased, incarcerated, out of jurisdiction) prior to review by the clinics. Lists of potentially randomizable patients were sent back to the DPH by secure file transfer. After de-identification, the list of patients was sent to Yale for randomization. Data from the clinics and eHARS were coded using R version 3.5.1 before exporting to REDCap, a secure web-based electronic data capture tool hosted at Yale University [22–27]. Patients were stratified by clinic and randomized 1:1 to the Disease Intervention Specialist (DIS) or standard of care (SOC).
(TIFF)

## Acknowledgments

We acknowledge the contributions of the clinics, the DIS workers and supervisors and other data personnel at the CT DPH.

## Author Contributions

**Conceptualization:** Merceditas Villanueva, Janet Miceli, Lisa Nichols, Frederick Altice.

**Data curation:** Janet Miceli, Suzanne Speers, Constance Carroll.

**Formal analysis:** Merceditas Villanueva, Janet Miceli, Constance Carroll.

**Funding acquisition:** Merceditas Villanueva, Heidi Jenkins.

**Investigation:** Merceditas Villanueva, Frederick Altice.

**Methodology:** Merceditas Villanueva, Janet Miceli, Suzanne Speers, Frederick Altice.

**Project administration:** Merceditas Villanueva, Lisa Nichols, Heidi Jenkins.

**Supervision:** Lisa Nichols, Heidi Jenkins, Frederick Altice.

**Validation:** Constance Carroll.

**Writing – original draft:** Merceditas Villanueva.

**Writing – review & editing:** Merceditas Villanueva, Janet Miceli, Suzanne Speers, Frederick Altice.

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
