## [Decision Letter · Decision Letter 0]

21 Feb 2022

PONE-D-21-27478Advancing data to care strategies for persons with HIV using an innovative reconciliation processPLOS ONE

Dear Dr. Villanueva,

Thank you for submitting your manuscript to PLOS ONE. After careful consideration, we feel that it has merit but does not fully meet PLOS ONE’s publication criteria as it currently stands. Therefore, we invite you to submit a revised version of the manuscript that addresses the points raised during the review process. Please submit your revised manuscript by Apr 07 2022 11:59PM. If you will need more time than this to complete your revisions, please reply to this message or contact the journal office at plosone@plos.org. Please include the following items when submitting your revised manuscript:A rebuttal letter that responds to each point raised by the academic editor and reviewer(s). You should upload this letter as a separate file labeled 'Response to Reviewers'.A marked-up copy of your manuscript that highlights changes made to the original version. You should upload this as a separate file labeled 'Revised Manuscript with Track Changes'.An unmarked version of your revised paper without tracked changes. You should upload this as a separate file labeled 'Manuscript'.

We look forward to receiving your revised manuscript.

Kind regards,

Sungwoo Lim, DrPH

Academic Editor

PLOS ONE

https://journals.plos.org/plosone/s/fileid=ba62/PLOSOne_formatting_sample_title_authors_affiliations.pdf".

Reviewers' comments:

Reviewer's Responses to Questions

**Comments to the Author**

1. Is the manuscript technically sound, and do the data support the conclusions?

Reviewer #1: Yes

Reviewer #2: Partly

2. Has the statistical analysis been performed appropriately and rigorously? 

Reviewer #1: Yes

Reviewer #2: I Don't Know

3. Have the authors made all data underlying the findings in their manuscript fully available?

Reviewer #1: Yes

Reviewer #2: Yes

4. Is the manuscript presented in an intelligible fashion and written in standard English?

Reviewer #1: Yes

Reviewer #2: Yes

5. Review Comments to the Author

Reviewer #1: This is an excellent manuscript that meets all of the criteria for publication in PLOS ONE. I think it is an important and interesting contribution to the literature. It addresses an important topic, the methods are well described, the results are clear, and the conclusions are sound and based on the data presented. Thank you for the opportunity to review this excellent paper!

Reviewer #2: The issue of out of care is certainly of paramount importance in efforts to end the HIV epidemic in the United States and globally. The authors are commended for their work in this area.

Figure 1 and accompanying text: The reference to boxes was quite vexing as there were not separate boxes but lines in the same box for Boxes B-D and Box A was not labeled as Box A. This needs clarification as it took me some time to sort out this issue.

One piece of information that I did not find included and is critical to understand weigh the validity of the conclusions is the timeframe for the "Well Patient"- two negative viral loads at least 6 months apart. From my reading of the paper, the last two available viral loads would be used for this measure. How long ago was that? I think it would be helpful to have the range and median available for the viral load measure throughout the paper (in the text or tables) for the reader to guage the recency of those values.

The authors use relatively old CDC estimates for out of care. Is that because that estimate correlates with the timing of the study? If so, please clarify in the text.

There appears to be a typographical error in page 4, paragraph 4, line 8 "Formuse of HD staff"

6. PLOS authors have the option to publish the peer review history of their article (what does this mean?). If published, this will include your full peer review and any attached files.

Reviewer #1: No

Reviewer #2: No

---

## [Author Response · Author response to Decision Letter 0]

5 Apr 2022

Dear editors:

We thank the reviewers for their comments on our manuscript. Specific responses to individual comments are detailed below:

Reviewer #1: This is an excellent manuscript that meets all of the criteria for publication in PLOS ONE. I think it is an important and interesting contribution to the literature. It addresses an important topic, the methods are well described, the results are clear, and the conclusions are sound and based on the data presented. Thank you for the opportunity to review this excellent paper!

We thank the reviewer for endorsing the importance of our study.

Reviewer #2: The issue of out of care is certainly of paramount importance in efforts to end the HIV epidemic in the United States and globally. The authors are commended for their work in this area.

We thank the reviewer for this comment.

Figure 1 and accompanying text: The reference to boxes was quite vexing as there were not separate boxes but lines in the same box for Boxes B-D and Box A was not labeled as Box A. This needs clarification as it took me some time to sort out this issue.

Thank you for requesting this clarification. We have revised Figure 1. Flowchart for Enrollment of Patients Who Disengage from Medical Care to specifically delineate Box A and also to separate out Boxes B, C, D. 

One piece of information that I did not find included and is critical to understand weigh the validity of the conclusions is the timeframe for the "Well Patient"- two negative viral loads at least 6 months apart. From my reading of the paper, the last two available viral loads would be used for this measure. How long ago was that? I think it would be helpful to have the range and median available for the viral load measure throughout the paper (in the text or tables) for the reader to guage the recency of those values.

Thanks for this clarification as the recency of VL testing is important to assessing clinical status. In the Methods section, we added details to specify that the designation of “Well Patient” required that at the time of case conference (encompassing the prior 18 months which included the 12 month “in care” period and a 6 month “out of care” period), there were 2 VLs at least 6 months apart that were <20 copies/ml AND there were no detectable VLs during the in-care period. So all “Well Patients” had relevant test results within at most an 18 month timeframe. Most “Well Patients” had available VL results within the prior 6-12 months. 

The authors use relatively old CDC estimates for out of care. Is that because that estimate correlates with the timing of the study? If so, please clarify in the text.

We appreciate this comment. We originally quoted the older CDC estimates of out-of-care from 2016 which estimated that 49% of PWH are retained in care and 53% are virally suppressed. Our study was conducted from 11/2016-7/2018 which partially encompasses the earlier timeframe. However, we acknowledge that there were improvements in 2018 CDC estimates showing that 58% of PWH were retained in care and 65% were virally suppressed. We have modified the abstract and introduction to cite these improved estimates.

There appears to be a typographical error in page 4, paragraph 4, line 8 "Formuse of HD staff"

This typographical error has been corrected and now reads “use of HD staff to review surveillance records...”

Additional revisions:

Additional edits to the Methods section were made to include a full ethics statement.

Re: Data Availability: All our data are-de-identified and open access. Anyone who wants to review the data can provide a data request to receive data for analysis after it has been reviewed by our publication committee. 

We look forward to your consideration of our revised manuscript.

Sincerely,

Merceditas S. Villanueva, MD

---

## [Decision Letter · Decision Letter 1]

19 Apr 2022

Advancing data to care strategies for persons with HIV using an innovative reconciliation process

PONE-D-21-27478R1

Dear Dr. Villanueva,

We’re pleased to inform you that your manuscript has been judged scientifically suitable for publication and will be formally accepted for publication once it meets all outstanding technical requirements.

Kind regards,

Sungwoo Lim, DrPH

Academic Editor

PLOS ONE

Reviewers' comments:

Reviewer's Responses to Questions

**Comments to the Author**

1. If the authors have adequately addressed your comments raised in a previous round of review and you feel that this manuscript is now acceptable for publication, you may indicate that here to bypass the “Comments to the Author” section, enter your conflict of interest statement in the “Confidential to Editor” section, and submit your "Accept" recommendation.

Reviewer #1: All comments have been addressed

Reviewer #2: All comments have been addressed

2. Is the manuscript technically sound, and do the data support the conclusions?

Reviewer #1: Yes

Reviewer #2: Yes

3. Has the statistical analysis been performed appropriately and rigorously? 

Reviewer #1: Yes

Reviewer #2: Yes

4. Have the authors made all data underlying the findings in their manuscript fully available?

Reviewer #1: Yes

Reviewer #2: Yes

5. Is the manuscript presented in an intelligible fashion and written in standard English?

Reviewer #1: Yes

Reviewer #2: Yes

6. Review Comments to the Author

Reviewer #1: The authors have satisfactorily addressed all reviewer comments.

The manuscript meets all criteria for publication in PLoS ONE.

Reviewer #2: All my comments have been addressed. This is an excellent paper and will be instructive as states work engage patients with HIV who are out of care.

7. PLOS authors have the option to publish the peer review history of their article (what does this mean?). If published, this will include your full peer review and any attached files.

Reviewer #1: No

Reviewer #2: No

---

## [Editor Report · Acceptance letter]

25 Apr 2022

PONE-D-21-27478R1 

Advancing data to care strategies for persons with HIV using an innovative reconciliation process 

Dear Dr. Villanueva:

I'm pleased to inform you that your manuscript has been deemed suitable for publication in PLOS ONE. Congratulations! Your manuscript is now with our production department. 

Kind regards, 

on behalf of

Dr. Sungwoo Lim 

Academic Editor

PLOS ONE